# Multiplicative Filter Networks

**Rizal Fathony**[*]
Bosch Center for Artificial Intelligence
Pittsburgh, PA
`rizal.fathony@us.bosch.com`

**Anit Kumar Sahu**[*†]
Amazon Alexa AI
Seattle, WA
`anit.sahu@gmail.com`

**Devin Willmott**[*]
Bosch Center for Artificial Intelligence
Pittsburgh, PA
`devin.willmott@us.bosch.com`

**J. Zico Kolter**
Bosch Center for Artificial Intelligence
Carnegie Mellon University
Pittsburgh, PA
`zkolter@cs.cmu.edu`

## Abstract

Although deep networks are typically used to approximate functions over high dimensional inputs, recent work has increased interest in neural networks as function approximators for low-dimensional-but-complex functions, such as representing images as a function of pixel coordinates, solving differential equations, or representing signed distance functions or neural radiance fields. Key to these recent successes has been the use of new elements such as sinusoidal nonlinearities or Fourier features in positional encodings, which vastly outperform simple ReLU networks. In this paper, we propose and empirically demonstrate that an arguably simpler class of function approximators can work just as well for such problems: multiplicative filter networks. In these networks, we avoid traditional compositional depth altogether, and simply multiply together (linear functions of) sinusoidal or Gabor wavelet functions applied to the input. This representation has the notable advantage that the entire function can simply be viewed as a linear function approximator over an exponential number of Fourier or Gabor basis functions, respectively. Despite this simplicity, when compared to recent approaches that use Fourier features with ReLU networks or sinusoidal activation networks, we show that these multiplicative filter networks largely outperform or match the performance of these approaches on the domains highlighted in these past works.

## 1 Introduction

Neural networks are most commonly used to approximate functions over high-dimensional input spaces, such as functions that operate on images or long text sequences. However, there has been a recent growing interest in neural networks used to approximate low-dimensional-but-complex functions: for example, one could represent a continuous image as a function $f : \mathbb{R}^2 \to \mathbb{R}^3$ where the input to this function specifies $(x, y)$ coordinates of a location in the image, and the output specifies the RGB value of the pixel at that location. However, two recent papers in particular have argued that specific architectural changes are required to make (fully-connected) deep networks suitable to this task: Sitzmann et al. (2020) employ sinusoidal activation functions within a multi-layer networks (called the SIREN architecture); and Tancik et al. (2020) propose random Fourier features input to a traditional ReLU-based network. Both papers show that the resulting networks can approximate these low-dimensional functions much better than simple feedforward ReLU networks, and achieve striking results in representing fairly complex functions (e.g. 3D signed distance fields or neural radiance fields) with a high degree of fidelity. However, the precise benefit of sinusoidal bases or a first layer of Fourier features seems difficult to characterize, and it remains unclear why such representations work well for these tasks.

---

[*]The first three authors contributed equally, listed in alphabetical order by last name.
[†]Work done while at Bosch Center for Artificial Intelligence.

In this paper, however, we argue and empirically demonstrate that an arguably simpler class of functions can work as well or better than these previously-proposed networks on this task. Specifically, we propose an architecture we call the *multiplicative filter network* (MFN). Unlike a traditional multi-layer network that achieves representation power through compositional depth, the MFN instead simply repeatedly applies nonlinear filters (such as a sinusoid or a Gabor wavelet function) to the network's input, then multiplies together linear functions of these features. The notable advantage of this representation that, owing to the multiplicative properties of Fourier and Gabor filters, the entire function is ultimately just a linear function of (an exponential number of) these Fourier or Gabor features of the input. Indeed, we can express the exact linear form of these MFNs, which can make their analysis considerably simpler than that for deep networks, where compositions of nonlinear activation's make the entire function difficult to characterize.

In this work, we show that despite this simplicity, the proposed networks often perform as well or better than the previously proposed SIREN or Fourier feature networks. Specifically, we compare our approach on networks with comparable numbers of parameters to the exact benchmarks proposed in the SIREN and Fourier features papers. We show that MFNs achieve better performance deltas when increasing the depth or width of the networks. Despite this, we do emphasize that SIREN networks, in particular, appear to retain some notable advantages over MFNs, such as a bias towards smoother regions in the represented function and its gradients. However, especially given the fact that MFNs ultimately just correspond to a linear Fourier or Wavelet representation of a low-dimensional function, we believe they should be considered a standard benchmark for future work on such problems, to indicate where the compositional depth of typical deep networks can propose a substantial benefit.

## 2 BACKGROUND AND RELATED WORK

Our approach is related to many previous works in Fourier and Wavelet transforms, random Fourier features, and implicit neural representations. We explore the connection among the areas below.

**Fourier and Wavelet transforms.** Transforming time or space domain signals to frequency domain using transforms such as Fourier and Wavelet transforms have been at the heart of many developments in image processing, signal processing, and computer vision. In particular, the Fourier transform (Bracewell & Bracewell, 1986; Vetterli et al., 2014) and its various forms have found usage in myriad applications, such as spectroscopy, quantum mechanics, signal processing. Wavelet transforms, which in particular aid in multi-scale analysis, have been found to be particularly useful in data compression, JPEG2000 (Rabbani, 2002) being one example.

**Random Fourier features.** A seminal work by Rahimi & Recht (2008) demonstrates the power of Fourier transform in machine learning applications. They show that simply projecting the original dataset into random Fourier bases vastly improves the expressiveness of models as it approximates kernel computations. Many subsequent works apply the Fourier features and variations (Rahimi & Recht, 2009; Le et al., 2013; Yu et al., 2016) to improve machine learning algorithm performance in many domain areas, including classification (Sun et al., 2018; Rawat et al., 2019), regression (Avron et al., 2017; Brault et al., 2016), clustering (Chitta et al., 2012; Liu et al., 2019), online learning (Lin et al., 2014; Hu et al., 2015), and deep learning (Xue et al., 2019; Mehrkanoon & Suykens, 2018; Rick Chang et al., 2016; Mairal et al., 2014; Jacot et al., 2018; Tancik et al., 2020).

**Implicit neural representations.** A recent line of work in representing signals as a continuous function parameterized by neural network (instead of using the traditional discrete representation) is gaining popularity. This strategy has been used to represent different objects such as images (Nguyen et al., 2015; Stanley, 2007), shapes (Park et al., 2019; Genova et al., 2019; Chen & Zhang, 2019; Chabra et al., 2020), scenes (Mildenhall et al., 2020; Sitzmann et al., 2019; Jiang et al., 2020; Niemeyer et al., 2020), and textures (Oechsle et al., 2019; Henzler et al., 2020). In most of these applications, the standard neural networks architecture with multi-layer perceptrons and ReLU activation function is often used. Recently, motivated by the success of Fourier transform in machine learning, a few papers have suggested architectural changes that integrate periodic nonlinearities into the network. Mildenhall et al. (2020); Zhong et al. (2020); Tancik et al. (2020) proposed the use of sinusoidal mapping of the input features (Rahimi & Recht, 2008) that uses positional encoding and Gaussian random distribution in the mapping. Others (Klocek et al., 2019; Sitzmann et al., 2020) have proposed the use of sinusoidal activation function within a multi-layer perceptron architecture.

Both of these strategies are demonstrated to vastly improve the results on many object representation tasks.

## 3 MULTIPLICATIVE FILTER NETWORKS

A traditional $k$-layer deep network $f : \mathbb{R}^n \to \mathbb{R}^m$ is typically defined by a recurrence such as:

$$z^{(1)} = x$$
$$z^{(i+1)} = \sigma\left(W^{(i)}z^{(i)} + b^{(i)}\right), \; i = 1, \ldots, k-1 \tag{1}$$
$$f(x) = W^{(k)}z^{(k)} + b^{(k)}$$

where $\sigma$ denotes a nonlinearity applied elementwise, $W^{(i)} \in \mathbb{R}^{d_{i+1} \times d_i}$ and $b^{(i)} \in \mathbb{R}^{d_{i+1}}$ denote the weight and bias of the $i$th layer, and $z^{(i)} \in \mathbb{R}^{d_i}$ denotes the hidden unit at layer $i$. We refer to these networks as compositional depth networks, because each nonlinearity is applied compositionally to outputs of the previous nonlinearity in order to achieve its representational complexity.

The SIREN or Fourier feature networks of (Sitzmann et al., 2020) and Tancik et al. (2020) respectively can be viewed as simple specializations of this structure. In a SIREN network, one uses the sinusoid $\sigma(x) = \sin(x)$ as the nonlinearity, plus proper initialization of the weights and scaling of the input. In a Fourier features network, one replaces the input layer with

$$z^{(1)} = \left[ \begin{array}{c} \sin(\Omega x + \phi) \\ \cos(\Omega x + \phi) \end{array} \right] \tag{2}$$

where $\Omega \in \mathbb{R}^{\frac{d_1}{2} \times n}$ is matrix of random $\mathcal{N}(0, \tau^2)$ variables ($\tau$ being a hyperparameter of the method), but with the typical ReLU nonlinearities $\sigma(x) = \text{ReLU}(x)$.

Our proposed *multiplicative filter network*, in contrast, uses a different recursion that never results in composition of nonlinear functions. Specifically, an MFN is defined via the following recursion

$$z^{(1)} = g\left(x; \theta^{(1)}\right)$$
$$z^{(i+1)} = \left(W^{(i)}z^{(i)} + b^{(i)}\right) \circ g\left(x; \theta^{(i+1)}\right), \; i = 1, \ldots, k-1 \tag{3}$$
$$f(x) = W^{(k)}z^{(k)} + b^{(k)}$$

where $\circ$ denotes elementwise multiplication, $W^{(i)}$, $b^{(i)}$, $z^{(i)}$ are all defined as above, but where $g : \mathbb{R}^n \to \mathbb{R}^{d_i}$ is parameterized by parameters $\theta^{(i)}$ (the size of $\theta^{(i)}$ can vary to implicitly define the output dimensions $d_i$) and denotes a *nonlinear filter* applied to the input directly. Of immediate importance here is that in such a network, we never apply a nonlinearity to the output of a previous nonlinearity. All the nonlinearity of the network occurs within the $g$ functions; layers $z^{(i)}$, after passing through a linear function, are simply multiplied by new filters of the input. This results in a considerably different type of function that is currently employed by most multi-layer networks, and indeed it is largely only by convention that we refer to such a function as a "network" at all.

We now present two instantiations of the MFN, using sinusoids or a Gabor wavelet as the filter $g$; we call these two networks the FOURIERNET and GABORNET respectively. As we show, the crucial property of a function $f$ represented by a FOURIERNET or GABORNET is that the entire function $f$ can also be written as a linear combination of sinusoids and Gabor wavelets of the input respectively (albeit an exponentially large number of such features, but of course also with a highly reduced space of allowable coefficients on this exponential number of terms, since there are only a polynomial number of parameters that define the MFN). Thus, we would claim that the MFN really looks more like a (rich) Fourier or Wavelet representation of the underlying signal, just one that happens to have a similar parameterization as deep networks (and which can be tuned by typical gradient descent methods).

### 3.1 MULTIPLICATIVE FOURIER NETWORKS

As our first instantiation of the MFN, we consider using a simple sinusoidal filter

$$g(x; \theta^{(i)}) = \sin(\omega^{(i)}x + \phi^{(i)}) \tag{4}$$

with parameters $\theta^{(i)} = \{\omega^{(i)} \in \mathbb{R}^{d_i \times n}, \phi^{(i)} \in \mathbb{R}^{d_i}\}$. We term such a network the FOURIERNET, as the sinusoidal activation (with arbitrary phase shifts, to represent sine or cosine functions equally) corresponds naturally to a Fourier random feature representation of the entire function.

An immediate and compelling feature of the FOURIERNET, compared to networks based upon composition, is that its output can be directly viewed as a linear function of (an exponential number of) Fourier bases, with a low-rank set of coefficients determined by the parameters of the network. This is conveyed by the following theorem.

**Theorem 1.** *The output of a Fourier Network is given by a linear combination of sinusoidal bases,*

$$f_j(x) = \sum_{t=1}^{T} \bar{\alpha}_t \sin(\bar{\omega}_t x + \bar{\phi}_t) + \bar{b}, \tag{5}$$

*for some coefficients $\bar{\alpha}_{1:T}$, frequencies $\bar{\omega}_{1:T}$, phase offsets $\bar{\phi}_{1:T}$, and bias term $\bar{b}$.*

In other words, the FOURIERNET represents its final function as a linear combination of traditional Fourier bases, just as do "classical" random Fourier features, for instance. The key element of the proof (given in the appendix) is the fact that for two Fourier filters, with parameters $\omega, \phi$ and $\tau, \psi$ respectively, their elementwise product can be transformed to a *sum* of the same type of filters

$$\sin(\omega x + \phi) \circ \sin(\tau x + \psi) = \frac{1}{2}\cos\left((\omega - \tau)x + \phi - \psi\right) - \frac{1}{2}\cos\left((\omega + \tau)x + \phi + \psi\right) \tag{6}$$

(note that the cosine can be expressed as a sine with a separate phase offset). Moreover, an inspection of the proof also lets us compute the exact coefficients of the linear expansion, as a function of the network parameters. This is shown in the following corollary.

**Corollary 1.** *Let $i_1, i_2, \ldots, i_{k-1}$ range over all $\prod_{j=1}^{k-1} d_j$ possible indices of each hidden unit of each layer of an MFN, and let $s_2, \ldots, s_k \in \{-1, +1\}$ range over all $2^{k-1}$ possible binary signs; then the expansion of $z_{i_k}^{(k)}$ from (2) is given by all the terms*

$$\begin{aligned}
\bar{\alpha} &= \left\{ \frac{1}{2^{k-1}} W_{i_k, i_{k-1}}^{(k-1)} \cdots W_{i_3, i_2}^{(2)} W_{i_2, i_1}^{(1)} \right\} \\
\bar{\omega} &= \left\{ s_k \omega_{i_k}^{(k)} + \ldots + s_2 \omega_{i_2}^{(2)} + \omega_{i_1}^{(1)} \right\} \\
\bar{\phi} &= \left\{ s_k \phi_{i_k}^{(k)} + \ldots + s_2 \phi_{i_2}^{(2)} + \phi_{i_1}^{(1)} + \frac{\pi}{2} \sum_{i=2}^{k} s_k \right\}.
\end{aligned} \tag{7}$$

*with a similar form for terms that begin at the $i > 1$ layer, multiplied by the corresponding $b_{i_j}^{(j)}$ term.*

This corollary follows simply by inspection of the proof in the appendix, noting that each additional multiplicative layer creates both a positive and negative combination of frequencies in the sinusoid terms, and a multiplication of the corresponding entries of $W$. In other words, the multiplicative "depth" of the FOURIERNET allows it to represent an exponential number of sinusoidal functions, but with the constraint that the actual number of coefficients on these features is given by a "low-rank" tensor consisting mainly of the coefficients in the $W$ matrices. This expansion also suggests a method for initializing parameters specific to this network in a manner that scales appropriately with the network size. Specifically, however $W^{(i)}$s are initialized (typically random uniform or Gaussian, though here with an additional scaling factor that depends on the relative scale of the input), one should divide these terms by $\sqrt{k}$, to ensure that the variance of the final frequency $\omega_t$ is independent of the number of layers.

## 3.2 MULTIPLICATIVE GABOR NETWORKS

A well-known deficiency of the pure Fourier bases is that they have global support, and thus may have difficulty representing more local features. A common alternative to these bases is the use of Gabor filter to capture both a frequency and spatial locality component. Specifically, we consider a Gabor filter of the form

$$g_j(x; \theta^{(i)}) = \exp\left(-\frac{\gamma_j^{(i)}}{2} \left\| x - \mu_j^{(i)} \right\|_2^2\right) \sin\left(\omega_j^{(i)} x + \phi_j^{(i)}\right) \tag{8}$$

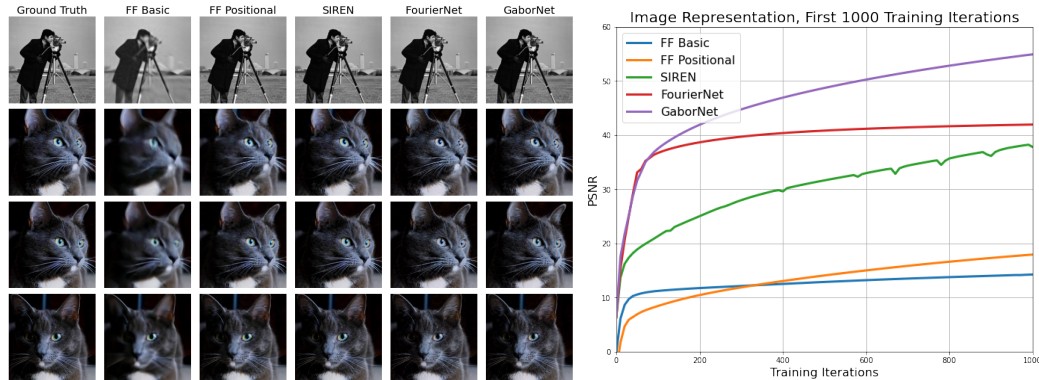

Figure 1: Left: Performance of various models on an image representation task (top row) and three frames from a video representation task (remaining rows). Leftmost column shows ground truth. Right: PSNR of each model in the image reconstruction task for the first 1000 training iterations.

with parameters $\theta^{(i)} = \left\{ \gamma^{(i)}_{1:d_i} \in \mathbb{R}, \mu^{(i)}_{1:d_i} \in \mathbb{R}^n, \omega^{(i)}_{1:d_i} \in \mathbb{R}^n, \phi^{(i)}_{1:d_i} \in \mathbb{R} \right\}$ (here $\mu^{(i)}_j$ denotes the mean of the $j$th Gabor filter and $\gamma^{(i)}_j$ denotes the scale term), and where for simplicity we specify the functional form of each $j = 1 \ldots, d_i$ coordinates of the function $g : \mathbb{R}^n \to \mathbb{R}^{d_i}$. We call the MFN using this filter the *GaborNet*.

As with the FourierNet, a compelling feature of the Gabor network is that the final function $f$ can be represented as a linear combination of Gabor filters. This is captured by the following theorem:

**Theorem 2.** *The output of a Gabor Network is given by a linear combination of Gabor bases,*

$$f_j(x) = \sum_{t=1}^{T} \bar{\alpha}_t \exp\left( -\frac{1}{2} \bar{\gamma}_t \| x - \bar{\mu}_t \|^2 \right) \sin(\bar{\omega}_t x + \bar{\phi}_t) + \bar{b}, \tag{9}$$

*for coefficients $\bar{\alpha}_{1:T}$, scales $\bar{\gamma}_{1:T}$, means $\bar{\mu}_{1:T}$, frequencies $\bar{\omega}_{1:T}$, phase offsets $\bar{\phi}_{1:T}$, and bias term $\bar{b}$.*

The proof is given in the appendix, but the basic procedure is the same as the above: using the fact that just like Fourier filters, the product of Gabor filters is also a linear combination of (a different set of) Gabor filters. Likewise, we can also compute the explicit form of the coefficients and for this linear basis expansion, with the explicit form again given in the appendix. One relevant point, though, is how we choose initializations for the $\gamma$ and $\mu$ parameters. Since $\gamma$ effectively acts as an inverse covariance term of a Gaussian, a $\mathrm{Gamma}(\alpha, \beta)$ random variable (the conjugate prior of the Gaussian inverse covariance), is a reasonable choice for this parameter. And since the $\bar{\gamma}$ functions in the final linear expansion end up being a *sum* of the individual random $\gamma^{(i)}$ terms at each layer, we scale each layer's $\alpha$ term by $1/k$ to effectively control this parameter at the final layer. We also simply choose each $\mu^{(i)}$ to be uniformly distributed over the range of the allowable input space $x$.

## 4 EXPERIMENTAL RESULTS

We test MFNs on a broad range of representation tasks, showing that the relative simplicity of MFNs improves upon the performance of existing neural representation methods. Our set of experiments draws from those presented in Sitzmann et al. (2020) alongside SIREN (image representation, shape representation, and differential equation experiments) and in Tancik et al. (2020) alongside Fourier feature networks with Gaussian random features, which we call FF Gaussian (image generalization and 3D inverse rendering experiments). In each case, we compare against the set of models tested in the original experiment (generally, either SIREN or FF Gaussian, along with a basic ReLU MLP). A PyTorch implementation of MFN is available at `https://github.com/boschresearch/multiplicative-filter-networks`, and full details on hyperparameters and training specifications are available in the appendix.

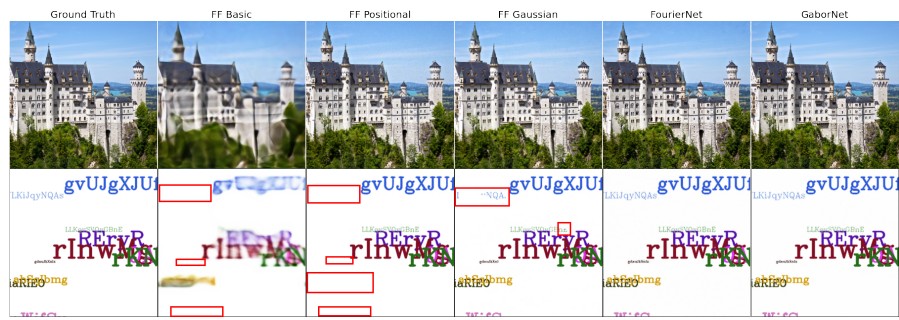

Figure 2: Image generalization samples from the *Natural* and *Text* datasets.

## 4.1 IMAGE REPRESENTATION & GENERALIZATION

We first examine the ability of several networks architectures in the task of image representation as described in Section 1, where we fit the network to a function $f : \mathbb{R}^2 \to \mathbb{R}^c$ using a dataset where input coordinates $(x, y)$ corresponding to the output pixel value at those coordinates (with $c = 1$ or $3$ for grayscale or RGB images, respectively). To demonstrate this, we construct such a dataset from a $256 \times 256$ pixel grayscale image and fit various models using a simple mean squared error (MSE) loss, including SIREN, GABORNET, FOURIERNET, and ReLU MLPs with and without positional encoding (PE). Visual results and a plot of PSNR early in training are shown in (Figure 1). In particular, both FOURIERNET and GABORNET show quicker initial convergence than other archi-

Table 1: PSNR of each model's reconstruction in image and video representation tasks after 10,000 training iterations. For video representation, mean ± standard deviation over all frames is reported.

| Method | PSNR (in dB) | |
| --- | --- | --- |
| | Image | Video |
| FF Basic | 20.13 | $24.09 \pm 1.03$ |
| FF Positional | 40.09 | $27.90 \pm 0.99$ |
| SIREN | 56.54 | $\mathbf{30.58 \pm 0.93}$ |
| FOURIERNET | 43.32 | $27.93 \pm 0.91$ |
| GABORNET | $\mathbf{73.98}$ | $29.83 \pm 0.71$ |

tectures. PSNRs after training (Table 1) show that SIREN eventually outperforms FOURIERNET, while GABORNET remains the best model throughout the training. Indeed, after only 1000 training iterations, GABORNET performs reconstruction better than all other models trained for 10 times longer.

We can broaden the task above to represent video by appending a third dimension to the input: the output corresponding to the input $(x, y, t)$ is the pixel value at $(x, y)$ at frame $t$. We aim to represent a 300 frame color video with $512 \times 512$ resolution in this manner, testing all of the architectures used in the previous experiment. As shown in Figure 1, SIREN and GABORNET are most capable of reproducing fine details of the original video, such as whiskers and and eye color. This is reflected in the PSNR of these reconstructions (Table 1); SIREN performs best with a PSNR over 30 dB, while the GABORNET reconstruction comes within 1 dB of SIREN and exhibits much lower variation across frames than any other model.

In addition to representing images, we demonstrate that the MFNs are able to generalize the representation to unseen pixels. We train the networks using only 25% of the image pixels (every other pixel in the width and height dimensions) and evaluate using the complete images. We compare the results of our methods with the Fourier feature networks on two datasets (natural and text images) presented in Tancik et al. (2020). The peak signal-to-noise ratio (PSNR) metric is used to evaluate the performance. As we can see from Table 2, both FOURIERNET and GABOR-

Table 2: Image generalization results (mean ± standard deviation of PSNR).

| Method | Natural | Text |
| --- | --- | --- |
| FF Basic | $21.61 \pm 2.62$ | $20.50 \pm 2.13$ |
| FF Positional | $25.13 \pm 4.01$ | $26.49 \pm 3.11$ |
| FF Gaussian | $25.57 \pm 4.18$ | $30.46 \pm 1.97$ |
| FOURIERNET | $26.03 \pm 2.77$ | $31.02 \pm 2.04$ |
| GABORNET | $\mathbf{26.18 \pm 2.95}$ | $\mathbf{31.19 \pm 2.00}$ |

NET outperform all versions of the Fourier feature networks that uses basic, positional encoding, and random Gaussian features. Some examples of the generated images are presented in Figure

Table 3: Total training loss after training for differential equations, where the iterations column refers to the number of iterations for which the networks were trained. PLI and PGI refer to Poisson equation based image reconstruction using Laplacians and gradients respectively. Helmholtz and wave refer to the single source inversion task.

| Problem | Iters | SIREN | FOURIERNET | GABORNET |
|---|---|---|---|---|
| PGI | 10000 | 0.55 | 1.16 | **0.46** |
| PLI | 10000 | 3432.56 | 76663.52 | **107.129** |
| Helmholtz | 50000 | 84949.92 | 46486.0 | **18845.75** |
| Wave | 10000 | 2090.15 | 691.34 | **441.13** |

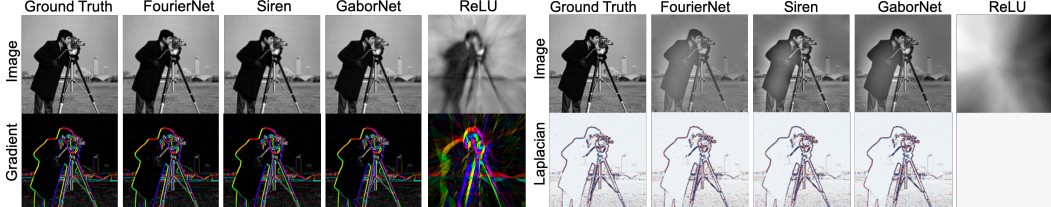

(a) Poisson Image Reconstruction: Gradients    (b) Poisson Image Reconstruction: Laplacians

Figure 3: Poisson Image Reconstruction: In the left and right figures, an image on the left is reconstructed using gradients and Laplacians respectively; the top row depicts the reconstructed images, while the bottom row indicates the fitted gradients and the fitted Laplacians.

2, with additional images in Appendix B.2. Visually, the MFNs' generated images also have better quality over the baselines, particularly in *Text* datasets. Some parts of the text are missing in the baselines images (highlighted with red rectangles in Figure 2), whereas the MFNs completely generate all parts of the text in the images.

## 4.2 DIFFERENTIAL EQUATIONS

In this section, we aim to solve boundary value problems which are supervised by different forms of gradient information from the functional at hand. We first focus on the Poisson equation, where we demonstrate image reconstruction in two settings where the supervision for the model is brought about by gradients and Laplacians respectively. It is worth noting that the model is never presented with real function values. We then focus on two 2nd order differential equations, namely, the Helmholtz equation and the wave equation, where we solve for the wave field, where the network is supervised by a known source function.

We demonstrate image reconstruction using gradients and compare the performance of FOURIER-NET and GABORNET with SIREN and ReLU MLP for the Poisson equation. We use the same loss function as in (Sitzmann et al., 2020) in (5). Figures 3a and 3b show that the image is reconstructed successfully when the networks are trained while being supervised by gradients and Laplacians respectively, while ReLU fails spectacularly. Table 3 depicts the losses of each method after 10000 iterations, where it can be seen that GABORNET beats other baselines in terms of performance.

The Helmholtz and wave equations are related to the physical modeling of diffusion and waves and are closely related to a Fourier transform. Hence, we focus our attention on describing the Helmholtz equation. We aim to solve for the wave field and compare the performance of FOURIERNET and GABORNET with SIREN and ReLU MLP. To accommodate for complex-valued solutions, the network is configured to output two values which can be interpreted as the real and imaginary parts. We use the same loss function as used in (Sitzmann et al., 2020) (see Section 4.3 of (Sitzmann et al., 2020) for details). Figure 4 shows the magnitude and the phase of the reconstructed wave front for a single Gaussian source placed at the center of a medium with uniform wave propagation velocity. Table 3 depicts the losses of each method after 50000 iterations, and shows that GABORNET beats FOURIERNET and SIREN in terms of performance, while, as previously shown in (Sitzmann et al., 2020), ReLU MLP fails miserably. Details pertaining to the network and training are relegated to Appendix B.3.

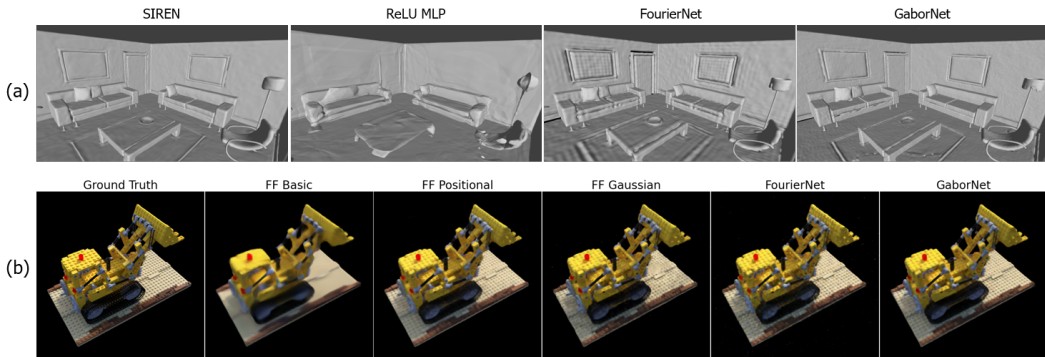

Figure 5: Shape representations from fitting signed distance functions (a). 2D rendered photographs from view synthesis experiments (b).

## 4.3 SHAPE REPRESENTATION VIA SIGNED DISTANCE FIELDS

Recent work (Park et al., 2019) has explored the problem of 3D shape representation with neural architectures, often with surprisingly effective results. This is done by training on raw geometric data: given an oriented point cloud, we seek to learn a function $f : \mathbb{R}^3 \to \mathbb{R}$ that takes points as input such that the zero level set $\{x \mid f(x) = 0\}$ of the network accurately represents the surfaces of the shape. Effective training objectives for this task has been explored in considerable depth; we use the training loss presented by Park et al. (2019) and used in Sitzmann et al. (2020), which includes terms involving the

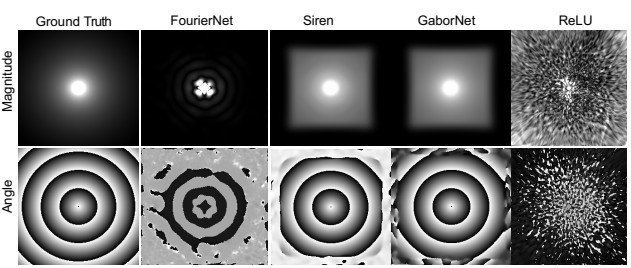

Figure 4: Solving Helmholtz equation for a single point source placed at the center of a medium with uniform wave propagation velocity. The top row presents the magnitude, while the bottom row presents the phase.

network's output (penalties to encourage SDF values near and away from 0 for surface and off-surface points, respectively) as well as its gradients (a term encouraging the gradient to match the surface points' normals, and a gradient norm penalty throughout the entire 3D space). A complete description of the loss function is left to the appendix.

Figure 5a shows the results of our shape representation task on SIREN, a standard ReLU MLP, and both variants of MFN. As can be seen, the ReLU network's fails to represent some features of the scene entirely, such as doorways, picture frames, and pillows. Both MFN architectures far outperform this baseline, and are able to reconstruct the room and objects within it to a recognizable degree. However, likely owing to its strong ability to produce smooth outputs and gradients, SIREN is largely able to avoid the visual artifacts on flat surfaces like walls that remain in reconstructions by FOURIERNET and, to a lesser extent, GABORNET.

## 4.4 3D INVERSE RENDERING FOR VIEW SYNTHESIS

In this view synthesis task, we aim to reconstruct 3D representation from the observed 2D photographs. Using the reconstructed 3D representation, we then render 2D images from new viewpoints. We use the "simplified Neural Radiance Fields (NeRF)" task on *Lego* dataset presented in Tancik et al. (2020). The networks are trained to predict the color (in RGB format) and the volume density at a given 3D location of the viewpoint. Volumetric rendering is then used to re-render the 2D image photograph at the viewpoint. The training loss is computed as the mean squared error between the rendered photograph and the actual 2D image observations. The results in Table 4 show that both MFNs perform competitively in the task. GABORNET has a slight advantage over

the baselines and FOURIERNET in terms of the overall PSNR metric. It also arguably produces a slightly better image rendering quality as shown in Figure 5b and other images in Appendix B.5.

## 5 CONCLUSION

We have introduced multiplicative filter networks (MFNs), a class of neural representation architectures that forego the usual compositional notion of network depth in favor of a similarly expressive multiplicative operation. They also admit a natural signal processing interpretation, as in the two instantiations of MFNs, FOURIERNET and GABORNET, which are proven to be exactly equivalent to a linear combination of sinusoidal or Gabor wavelet bases, respectively. In experiments, we show that, despite their simplicity relative to other deep architectures designed for implicit representation, MFNs stand up to or surpass the previous state of the art on a battery of representation tasks.

Table 4: View synthesis results (mean $\pm$ st.d. of PSNR).

| Method | PSNR |
|---|---|
| FF Basic | $23.37 \pm 0.96$ |
| FF Positional | $25.76 \pm 0.79$ |
| FF Gaussian | $25.76 \pm 0.92$ |
| FOURIERNET | $25.20 \pm 0.72$ |
| GABORNET | $\mathbf{25.81 \pm 0.76}$ |

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
