# OpenReview forum: "Multiplicative Filter Networks"
_ICLR.cc/2021/Conference — ICLR 2021 Poster_

### Official Review · AnonReviewer4 · 2020-10-17
**Incremental work with better performance than previous work**

**Rating:** 6
**Confidence:** 4

**Review:**

This paper propose a new network architecture, multiplicative filter network (MFN),  for implicit function reconstruction. The proposed network incorporate elementwise multiplication of nonlinear filters to replace to conventional nonlinear functions, e.g. ReLU, for better signal reconstruction. Two different multiplicative filter network is studied, FourierNet and GaborNet, which used Fourier and Gabor functions as the nonlinear functions in the MFN. Experiments have shown that the proposed method can outperform SIREN under the similar level of computational complexity.

Positive:
The paper is easy to follow and the proposed method seems to be working well especially for the GaborNet.

Negative:
On top of SIREN, this paper looks incremental. Although it makes some theoretical study that the proposed FourierNet and GaborNet can favorably reconstruct original signals, there is no theoretical comparisons with this work and SIREN. There is only empirical study to show that the results of the proposed method is better than SIREN (GaborNet only, the FourierNet performs worse than SIREN).

Open Question:
I have checked the SIREN and its source codes. Although SIREN can reconstruct original image favorably, its network structure is unfriendly for image editing and/or other image processing tasks. Have the authors of this paper encounter similar problem in MFN?

---

> ### Author Response · Authors · 2020-11-13
> **Thank you for your thoughtful review**
>
> Thank you for your thoughtful review. We now address the main concern below.
>
> - One of the major benefits of our method compared to the SIREN architecture is that we can easily characterize the entire function of the MFNs. We show that the entire function of FourierNet and GaborNet can simply be viewed as a linear function approximator over an exponential number of Fourier or Gabor basis functions, respectively. This establishes a connection of the network architecture with the traditional Fourier and Gabor wavelet transform, which are extensively studied in the literature and widely used in many application domains.
> The same connection is not present in SIREN: the compositional nature of SIREN makes it difficult to characterize in a signal processing context, which in turn makes it unclear why such representation works well for modeling low-dimensional-but-complex functions. In light of this, we did not see opportunities for a direct theoretical comparison of MFNs and SIREN. Could you elaborate on the kind of theoretical comparison you were hoping to see that was not present in the paper?
>
> - Could you please clarify the tasks in the image editing / other image processing tasks? Which part of the SIREN architecture makes it unfriendly for those tasks?

---

### Official Review · AnonReviewer3 · 2020-10-26
**novel proposal to replace conventional compositional networks to learn better representations, showing some marginal quantitative improvements over state-of-the-art, generalization capability remains a question.**

**Rating:** 6
**Confidence:** 3

**Review:**

**Summary**
This paper proposes two schemes to learn better representation, one based on Fourier features and another on Gabor filters. The idea is to place nonlinearity into the feature encoding rather than network layers like ReLU, and the goal is to learn better representation with more compact models. Preliminary results outperform existing techniques on several image and video representation tasks. One generalization is shown for image completion.

**Strength**
The idea is simple but seems effective. It's interesting to leverage Gabor filters' capability in spatial and frequency support to enable better feature encoding. Results show a better reconstruction of high frequency details and experiments are comprehensive covering multiple aspects of image and video tasks.

**Weakness**

    - I do not see how MFN 'largely outperforms' existing baseline methods. It is difficult to identify the quality difference between output from the proposed method and SIREN -- shape representation seems to even prefer SIREN's results (what is the ground truth for Figure 5a). The paper is based on the idea of replacing compositional models with recursive, multiplicative ones, though neither the theory nor the results are convincing to prove this linear approximation is better. I have a hard time getting the intuition of the advantages of the proposed method.
    - this paper, and like other baselines (e.g. SIREN) do not comment much on the generalization power of these encoding schemes. Apart from image completion, are there other experiments showing the non-overfitting results, for example, on shape representation or 3D tasks?
    - the proposed model has shown to be more efficient in training, and I assume it is also more compact in size, but there is no analysis or comments on that?

**Suggestions**

    - Result figures are hard to spot differences against baselines. It's recommended to use a zoom or plot the difference image to show the difference.
    - typo in Corollary 2 -- do you mean linear combination of Gabor bases?
    - It's recommended to add reference next to baseline names in tables (e.g. place citation next to 'FF Positional' if that refers a paper method)
    - In Corollary 1, $\Omega$ is not explicitly defined (though it's not hard to infer what it means).

---

> ### Author Response · Authors · 2020-11-13
> **Thank you for your thoughtful review.**
>
> Thank you for your thoughtful review. We now address the main concern below.
>
> - While we acknowledge that the improvement of MFNs over SIREN and Fourier features networks is slight in some experiments, we would contend that we do make some notable improvement in, for example, the image generalization tasks, specifically on the text dataset. However, the main contribution of the paper is providing a connection between the network architecture with the traditional Fourier and Gabor wavelet transform, which are extensively studied in the literature and widely used in many application domains. We show that the entire function of FourierNet and GaborNet can simply be viewed as a linear function approximator over an exponential number of Fourier or Gabor basis functions, respectively. In contrast, there is no characterization of the property of the SIREN and Fourier features networks, which makes it unclear why such representation works well for modeling low-dimensional-but-complex functions.
> The simplicity of the MFNs, where we can express the exact linear form of the entire function, makes it easier to characterize the network and opens up opportunities for further theoretical and empirical studies on similar network architectures.
>
> - In terms of the generalization power of the encoding schemes, we have two experiments that demonstrate the capability of our networks. First, in the image generalization task (Section 4.1) where the networks need to predict unseen pixels. The second experiment is the 3D inverse rendering task (Section 4.4). In this task, the networks need to reconstruct a 3D representation of an object (in this case, Lego blocks) from the observed 2d photographs from different viewpoints. From their learned 3d representation, the networks then need to predict 2d photographs from unseen viewpoints.
> In both of these tasks, we showed that the GaborNet outperforms the baselines approach that uses Fourier feature networks, whereas the FourierNet produces competitive results. These tasks also show some generalization capabilities of our MFNs.
>
> - In terms of the size of the networks, the MFNs have roughly similar size compared with the SIREN and Fourier feature networks. Given the similar architecture, each of those networks needs to store the same amount of parameters of the inner linear layers. GaborNet needs to learn additional parameters of \mu and \gamma, but their sizes are negligible compared with the size of parameters in the inner linear layers.
>
> - Thanks for the suggestion in the image comparison. We will add visual highlights on the main difference among the compared images to the appendix.
>
> - Yes, in Corollary 2, we mean “linear combination of Gabor bases”.
>
> - The current citation format (author year) makes it difficult to add a citation to the table. However, we will make sure that we include the citation for every baseline method at the beginning of the experiment section.
>
> - In the paragraph after Corollary 1, \Omega^{i} should be W^{i}. Thanks for pointing this out.

---

### Official Review · AnonReviewer2 · 2020-10-28
**Good paper, but some missing analysis**

**Rating:** 8
**Confidence:** 3

**Review:**

Summary.
The paper proposes a novel neural architecture for representing complex functions over on a low-dimensional domain (e.g. images on coordinates). Instead of following standard deep-learning ideas, the method stacks linear layers, but modulates each layer with a non-linear function (e.g. Fourier or Gabor functions). It is shown that this results in a linear combination of exponentially many Fourier/Gabor functions. Extensive experiments are performed, following the complete suite of experiments in SIREN [Sitzmann 2020], and it is shown that the proposed method outperforms available neural methods, except on video representation and signed-distance functions.

Reasons for score.
The paper proposes a novel method and shows superior performance on some tasks. However, generalization performance comparison for standard signal processing is missing. Hence, I suggest a weak accept.

Pro.
- simple and novel idea that fits nicely into standard frameworks
- experiments demonstrate its superiority on several tasks, even generalization compared to Fourier Features
- well written and easy to understand

Con.
- although theoretically shown, it remains empirically unclear if the method is able to leverage the exponential number of functions. I.e. an experiment to compare with standard Fourier and Gabor transformation should be performed. As a Fourier basis can represent any finitely sampled signal perfectly, a generalization analysis should be performed akin to pg.6 bottom. Put another way: when is the neuronal representation useful as opposed to just useing a Fourier/Gabor transform?
- it should be worked out more carfully what's novel throughout the paper. E.g. in section 4.2. "while ReLU fails spectacularly" - this has already been shown in the SIREN paper, and needs emphasis for readers not aware of this.


Minor.
- pg. 3 top: Wˆ{(i)} the dimensionality should be transposed, no?
- do all the methods in Figure 1 have the same numbers of free parameters?
- Table 3: is the relative performance stable wrt. to the number of iters? would be good to see the whole plot of performance vs. iters for all methods in the appendix.
- Figure 4: Ground -> Ground-truth; why is there a square box for SIREN and GaborNet? (it's not in the SIREN paper)


Udpate: with the additional experiments and corrections in the paper, I believe the paper is in good shape and contributes to the literature in the field. Hence it should be accepted.

---

> ### Author Response · Authors · 2020-11-13
> **Thank you for the thoughtful review**
>
> Thank you for the thoughtful review. We have responded to each of your points below.
> - Regarding when using MFNs is preferable to simply using Fourier/Gabor features: Thanks for making this point - we agree that an experiment that explores the benefits of using our network vs Fourier or Gabor features would strengthen the paper. Could you provide more details on what specifically you would like to see in such an experiment? For example, we are imagining that performing the image generalization task directly using random Fourier or Gabor features (perhaps, for a direct comparison, a number of features equal to the number of parameters in the W matrices of the MFNs we use in this experiment) would address this, but it would be useful to know if you have other ideas in mind for the experiment.
> - We will add wording to clarify which experimental results are novel in this paper, as opposed to those findings that were first shown in the SIREN paper.
> - In the models trained and tested on the image representation task shown in Figure 1, the total number of free parameters varies slightly; FourierNet and GaborNet have slightly more than other models, due to the additional learnable parameters in the input layers and filter function (i.e. \mu and \gamma in GaborNet). But these additional parameters are of order d_i (hidden size in layer i), whereas the total number of learnable parameters is dominated by the d_i^2 parameters in the hidden-to-hidden W matrices. Thus, we note that the favorable convergence of GaborNet and FourierNet in Figure 1 is despite a slightly larger number of parameters overall.
> - Yes, thank you for pointing this out - we intended to define the dimensions of W in the reverse order (i.e., as a d_{i+1} x d_i matrix). We will fix this.
> - In general, the convergence of the models we tested is fairly stable over time on these experiments, but we will add plots of performance over time for several of these tasks in the appendix.
> - We will fix the omitted label in Figure 4. Regarding the SIREN results in this figure, these results (and all of the experiments on SIREN models in our paper) come from experiments that we ran using the SIREN implementation the authors have provided. In general we have tried to be faithful to the original training hyperparameters in those experiments, but there may have been some minor differences from the original code; most notably, we reduced the batch size in several experiments to account for smaller GPUs. More details about our training setup are also available in the supplementary material.

---

> > ### Comment · AnonReviewer2 · 2020-11-17
> > **Fourier / Gabor Basis experiment**
> >
> > Thank you for your response.
> >
> > Concerning the experiment: what I had in mind is very similar to what you suggested.
> > Holding the number of parameters equal might be difficult, as for a linear basis the number of free parameters is the dimensionality of the space - i.e. the number of pixels (times 3 for RGB) in an image. Hence, the simplest experiment might be to use e.g. the fourier basis for upsampling of the signal (in the upsampled image, the low-frequency components stay the ones from the original sampling - higher frequencies are just 0), and then compare this standard image processing method with your results.

---

### Official Review · AnonReviewer1 · 2020-10-30
**Novel formulation using multiplicative filters on function approximation for low-dimensional-but complex functions**

**Rating:** 9
**Confidence:** 4

**Review:**

Summary:   This paper complements a class of recent work on function approximation for image function representation that leverages random fourier functions and sinusiodal activation functions (SIREN).  The main insight in the paper an alternative scheme that just repeatedly applies nonlinear filters (sinusoids, gabor wavelet functions) to the networks input and multiplies together linear functions of these features.  The authors show that due to multiplicative properties of fourier/ gabor filters the end resulting mapping is also represented by a linear combination of fourier/gabor filters. Essentially the entire function is a linear function approximator over an exponential number of fourier/gabor basis functions with a low-rank, polynomial number of coefficients of the MFN network.  Experiments are done to compare the proposed FourierNet, GaborNets against past work (FF, FF with positional encoding, FF Gaussian SIREN).  The results are competitive with other methods and in certain cases GaborNets outperform in PSNR benchmarks for various problems.

Pros:  I like the paper very much, it is very well written and clear. The idea is novel.

Cons: The empirical benchmarks are the same as used in the SIREN paper.  I would liked to see a discussion or analysis on tradeoffs between using SIREN or GaborNet (i.e. in what settings they are appropriate and perform perturbation analysis).

---

> ### Author Response · Authors · 2020-11-13
> **Discussion on tradeoffs between GaborNet and SIREN**
>
> Thank you for the thoughtful and positive review.
>
> In terms of the comparison between GaborNet and SIREN, our experiments show that overall, GaborNet provides a slight performance improvement compared to SIREN, particularly in the image representation, Poisson image reconstructions, and solving Helmholtz equation tasks (Section 4.1 and 4.2). Despite this, SIREN does appear to retain some notable advantages over MFNs, such as a bias towards smoother regions in the represented function and its gradients. We can see it from the shape representation experiment (Section 4.3), where SIREN has a notable benefit in representing smooth surfaces.
> From the theoretical perspective, GaborNet provides a key benefit over SIREN in terms of the characteristics of the entire function represented by the network. GaborNet can simply be viewed as a linear function approximator over an exponential number of Gabor basis functions. This simple and nice property of GaborNet is missing in the SIREN networks.

---

### Author Response · Authors · 2020-11-23
**Summary of our revision**

Dear respected reviewers.

We have submitted a revision to our paper. Below are the highlights of the revision:
- We fixed all the typos and wording problems, including:
  * We fixed the W notation in the first paragraph of page 3 to W^{i} \in R^{d_{i+1} x d_i}. This aligns with the usage of W in Eq (3), Eq (7), and other equations in the appendix.
  * The weight initialization in the paragraph after Corollary 1 should be W instead of \Omega. We fixed this.
  * We changed the image title in Figure 4 from ‘Ground’ to ‘Ground Truth’
  * We fixed the text in Theorem 2 to ‘linear combination of Gabor bases’ rather than ‘Fourier bases’
- We add a note in the discussion of Section 4.2 experiment to clarify that the failure of ReLU networks has been observed previously by SIREN paper (Sitzmann et al., 2020). We focus more on the comparison between GaborNet and SIREN.
- As requested by the reviewers, we added plots of performance vs iterations in the training progress to the appendix.
  * In the image representation task (Figure 1), both FourierNet and GaborNet show quicker initial convergences compared to other architectures. Eventually, the final performance shows GaborNet performs the best, followed by SIREN and FourierNet.
  * Similarly, in the video representation task (Figure 6), FourierNet and GaborNet show quicker initial convergences. But eventually, SIREN overtook both of them and maintained the best final performance, followed by GaborNet and FourierNet.
  * In the shape representation task (Figure 9), Both GaborNet and SIREN perform very similarly during the whole training progress. They both outperform FourierNet.
- We added visual highlights on the main differences among the compared images to the appendix, particularly for the image generalization task where the differences are clearly visible. Figure 8 now shows the highlighted differences.
- We added a discussion on the comparison between the proposed approach and the traditional FFT upsampling method in the appendix (Section B.2). As requested by the reviewers, we ran image upsampling on the image generalization task using the ‘Natural’ and ‘Text’ dataset. As expected, the traditional FFT upsampling method performs quite well on both datasets (73.80±12.20 and 66.78±8.14 PSNRs, respectively). However, this standard approach also has disadvantages compared to the implicit neural representations approach (the MFNs, SIRENs, and Fourier feature networks). The standard Fourier analysis does not scale well to higher dimensions, or it does not admit to backpropagation through more complex loss functions (e.g., in neural rendering fields). The lack of scalability and non-differentiability hinders the usage of FFT based upsampling methods in an end-to-end manner and hence adds significant barriers to such methods being employed in generic tasks. In contrast, the MFNs (and other implicit neural representations) provides generic methods that apply to generic tasks where we can learn based upon derivatives, as shown in our paper and the related works (SIREN and Fourier feature networks paper).

---

### Comment · ~Heng_Guo3 · 2023-07-27
**Question about Eq. (7)**

I have tried to write an analytical form of a two-layer MFN following Eq. (7). However, the result seems incorrect.
The key is that we cannot change the order of the dot product and element-wise operator, i.e.,
$\bf{A}\bf{b}  \odot   \bf{c} \neq \bf{A} (\bf{b}  \odot \bf{c}) \neq \bf{A}\bf{c}  \odot \bf{b}$

Consider a simple two-layer MFN network:
\begin{equation}
\bf{z}_0 = \sin(\bf{F}_0 \bf{x} + \bf{k}_0)
\end{equation}
\begin{equation}
\bf{z}_1 = \left(\bf{W}_0(\sin(\bf{F}_0 \bf{x} + \bf{k}_0)) + \bf{b}_0\right) \odot \sin(\bf{F}_1 \bf{x} + \bf{k}_1) =\bf{W}_0\sin(\bf{F}_0 \bf{x} + \bf{k}_0) \odot \sin(\bf{F}_1 \bf{x} + \bf{k}_1) + \bf{b}_0 \odot \sin(\bf{F}_1 \bf{x} + \bf{k}_1)
\end{equation}
\begin{equation}
\bf{z}_2 = \bf{W}_1 \left(  \bf{W}_0\sin(\bf{F}_0 \bf{x} + \bf{k}_0) \odot \sin(\bf{F}_1 \bf{x} + \bf{k}_1) + \bf{b}_0 \odot \sin(\bf{F}_1 \bf{x} + \bf{k}_1)       \right) \odot \sin(\bf{F}_2 \bf{x} + \bf{k}_2) + \bf{b}_1 \odot \sin(\bf{F}_2 \bf{x} + \bf{k}_2)
\end{equation}

If we want to expend the expression of $\bf{z}_2$ to the format of Eq. 7 of the main paper, we need to extract the term of $\bf{W}_1 \bf{W}_2$ and $\sin(\bf{F}_0 \bf{x} + \bf{k}_0) \odot \sin(\bf{F}_1 \bf{x} + \bf{k}_1)$. However, as mentioned above, we cannot apply the associative law in an expression containing both element-wise product and dot-product. I think we cannot simply write
\begin{equation}
\bf{\hat{z}}_2 = \bf{W}_1 \bf{W}_0\sin(\bf{F}_0 \bf{x} + \bf{k}_0) \odot \sin(\bf{F}_1 \bf{x} + \bf{k}_1) \odot \sin(\bf{F}_2 \bf{x} + \bf{k}_2)  + \bf{b}_0 \odot \sin(\bf{F}_1 \bf{x} + \bf{k}_1)  \odot \sin(\bf{F}_2 \bf{x} + \bf{k}_2) + \bf{b}_1 \odot \sin(\bf{F}_2 \bf{x} + \bf{k}_2)
\end{equation}
I have verified that $\bf{\hat{z}}_2 \neq \bf{z}_2$.
I wonder how we can simplify $\bf{z}_2$ in the format of Eq. (7). Am I missing something?

It would be nice if the authors can help to give a code to show a demo of analytical expression in Eq, (7), even with a two-layer MFN.

---

### Decision · Program_Chairs · 2021-01-07
**Final Decision**

**Decision:**

Accept (Poster)

**Comment:**

The paper proposes multiplicative filter networks (GaborNet and FourierNet) as functional approximations of deepnets. The proposed networks are a sequence of multiplications linear functions of sinusoidal or Gabor filters. The authors show that in some cases the performance of proposed networks outperforms the existing deepnets using ReLu activations. This representation is notably simpler as well. Moreover, compared to classical Fourier approach, the proposed method scales to higher dimensions in practice as well.
The downside of the paper is that it is not clear how to empirically use exponentially many Fourier functions. Moreover, proposed methods have more parameters, and the additional parameters are linear in size of the hidden layer.

The paper is clearly written and the authors improved the quality of the paper and added additional experiments to support their claim through the review process and I appreciate that.